# Antiskyrmions stabilized at interfaces by anisotropic Dzyaloshinskii-Moriya interactions

Markus Hoffmann [1], Bernd Zimmermann [1], Gideon P. Müller [1,2], Daniel Schürhoff[1], Nikolai S. Kiselev [1], Christof Melcher[3] & Stefan Blügel [1]

Chiral magnets are an emerging class of topological matter harboring localized and topologically protected vortex-like magnetic textures called skyrmions, which are currently under intense scrutiny as an entity for information storage and processing. Here, on the level of micromagnetics we rigorously show that chiral magnets can not only host skyrmions but also antiskyrmions as least energy configurations over all non-trivial homotopy classes. We derive practical criteria for their occurrence and coexistence with skyrmions that can be fulfilled by (110)-oriented interfaces depending on the electronic structure. Relating the electronic structure to an atomistic spin-lattice model by means of density functional calculations and minimizing the energy on a mesoscopic scale by applying spin-relaxation methods, we propose a double layer of Fe grown on a W(110) substrate as a practical example. We conjecture that ultra-thin magnetic films grown on semiconductor or heavy metal substrates with $C_{2v}$ symmetry are prototype classes of materials hosting magnetic antiskyrmions.

[1] Peter Grünberg Institut and Institute for Advanced Simulation, Forschungszentrum Jülich and JARA, 52425 Jülich, Germany. [2] Science Institute of the University of Iceland, VR-III, 107 Reykjavík, Iceland. [3] Department of Mathematics I & JARA FIT, RWTH Aachen University, 52056 Aachen, Germany. Correspondence and requests for materials should be addressed to M.H. (email: m.hoffmann@fz-juelich.de)

Chiral magnetic skyrmions[1] are currently the subject of intense scientific investigations. The topological protection of their magnetic structure and particle-like properties with a well-defined topological charge offer good conditions for skyrmions becoming information-carrying particles in the field of spintronics[2] and this explains the additional motivation of their current intensive investigation.

Skyrmions in chiral magnets may appear as isolated solitons or condensed in regular lattices. Their stability results from the Dzyaloshinskii–Moriya interaction (DMI)[3, 4], which breaks the chiral symmetry of the magnetic structure. The energy and size is determined by the competition between the Heisenberg, Dzyaloshinskii–Moriya, and Zeeman interaction together with the magnetic anisotropy energy (MAE), here expressed in terms of the spin-lattice model applied to an interface geometry

$$H = -\sum_{ij} J_{ij}\left(\mathbf{S}_i \cdot \mathbf{S}_j\right) - \sum_{ij} \mathbf{D}_{ij} \cdot \left(\mathbf{S}_i \times \mathbf{S}_j\right) \\ - \sum_i \mathbf{B} \cdot \mathbf{S}_i + \sum_i K_\perp (\mathbf{S}_i \cdot \widehat{\mathbf{e}}_z)^2 \tag{1}$$

with classical spin $\mathbf{S}$ of length one at atomic sites $i$, $j$ and the corresponding microscopic pair $J_{ij}$, $\mathbf{D}_{ij}$, and on-site $K_\perp$ interactions. The DMI results from the spin–orbit interaction and is only non-zero for solids lacking bulk ($\mathbf{r} \nleftrightarrow -\mathbf{r}$) or structure inversion symmetry ($z \nleftrightarrow -z$). For applications in spintronics[2, 5, 6] skyrmions stabilized in systems with surface- or interface-induced DMI seem to be more promising than bulk systems with DMI. For these systems the dipolar interaction between magnetic moments is typically of minor importance and is added as an additional on-site contribution to the MAE of the interface, $K_\perp$ ($K_\perp < 0$ means, the easy axis of the magnetization is normal to the interface plane, i.e., along $\widehat{\mathbf{e}}_z$ direction).

The first example of a topologically non-trivial structure stabilized by the interface DMI was revealed for an Fe monolayer on an Ir(111) substrate[7]. Subsequent incremental modifications, e.g., adding a monolayer Pd on top of the Fe layer, led to the observation of single skyrmions stablized by interface DMI[8, 9]. With respect to future applications, interfaces offer a great variety of options for optimizing and controlling magnetic parameters: variation of the interface composition[10], interface crystal symmetry[11], the film thickness[8], as well as the fabrication of interlayers[12] and multilayers[13].

Taking the micromagnetic view of the DMI, which is valid in the limit of slowly varying magnetic textures, with the prototypical examples of interfacial or cubic DMI,

$$D[\mathbf{m}(\nabla \cdot \mathbf{m}) - (\mathbf{m} \cdot \nabla)\mathbf{m}]_z \quad \text{or} \quad D\, \mathbf{m} \cdot (\nabla \times \mathbf{m}), \tag{2}$$

respectively, for $\mathbf{m} = \mathbf{m}(\mathbf{r})$ with $\mathbf{m}(\mathbf{R}_i) = \mathbf{S}_i$, it may not be surprising that the community focuses primarily on the realization of skyrmions ($Q = -1$) rather than antiskyrmions characterized by $Q = +1$, where

$$Q(\mathbf{m}) = \frac{1}{4\pi} \int_{\mathbb{R}^2} \mathbf{m} \cdot \left(\partial_x \mathbf{m} \times \partial_y \mathbf{m}\right) \mathrm{d}\mathbf{r} \tag{3}$$

is the topological charge or $\mathbb{S}^2$ winding number. The sign of $Q$ is consistent with the ferromagnetic background defined by the direction $\widehat{\mathbf{e}}_z$ of the Zeeman field. For skyrmionic configurations with a well-defined skyrmion center, it is instructive to take into account the index or $\mathbb{S}^1$ winding number

$$\nu = \frac{1}{2\pi} \oint_\Gamma \frac{m_x \mathrm{d}m_y - m_y \mathrm{d}m_x}{m_x^2 + m_y^2} \tag{4}$$

of the horizontal magnetization field $m_\parallel = (m_x, m_y)$, which

assumes the same integer value for every oriented Jordan curve $\Gamma$ enclosing the skyrmion core. In this case, $\nu$ can be considered as a secondary topological charge and the defining index to distinguish between skyrmion ($\nu = 1$) and antiskyrmion ($\nu = -1$) independently of the background state.

The classical skyrme problem exhibits a reflection symmetry, and the particle and antiparticle with the topological charges $Q = \pm 1$ exist with the same minimal energy[14–16]. This is different for chiral magnetic skyrmions, where this degeneracy is lifted as a consequence of chiral symmetry breaking. In the context of cubic DMI it has been shown that corresponding antiskyrmionic configurations have a strictly higher energy[17]. In fact, the densities in Eq. (2) can be mapped onto each other by a rigid (90 degree) rotation in horizontal spin space and can be considered equivalent. A close inspection of those proofs and simulations, which verify that the lowest-energy magnetization configuration of non-trivial topology is attained for $Q = -1$, reveals that a micromagnetic DMI Hamiltonian of the form (2) is assumed with a scalar DM constant $D$, also coined the spiralization.

Different versions of the DMI arise from rigid $O(2)$ transformations applied to the cubic DMI in Eq. (2). Some of which, e.g., those transforming to bulk crystals with $S_4$ or $D_{2d}$ symmetry, can lead to the stabilization of antiskyrmions rather than skyrmions, as has been recently demonstrated for an acentric tetragonal MnPtPdSn Heusler compound[18]. Due to symmetry restrictions, those $O(2)$ transformations are not available for the interfacial DMI (2) and thus antiskyrmions are absent in two-dimensional (2D) or film systems described by interfacial DMI with scalar spiralization. This picture has been confirmed recently by Koshibae and Nagaosa[19], who showed that antiskyrmions are unstable in 2D chiral magnets with scalar $D$, but may be realizable as bubbles in magnets with dipolar interactions.

Thus, more general and interesting are crystal symmetries where the spiralization constant $D$ is replaced by a generic tensor quantity $\mathcal{D}$, which is not included in the $O(2)$ orbit of Eq. (2). In this case of anisotropic DMI various scenarios of topological pattern formation, including non-symmetric skyrmions and antiskyrmions, are possible as shown by means of micromagnetic simulations for combined bulk and interface spin–orbit effects[20, 21]. A double layer Fe on W(110) exhibits a $C_{2v}$ symmetry and is an example with non-zero tensorial components of $\mathcal{D}$.

In the following we have extended our analysis[17] to 2D chiral systems governed by a micromagnetic energy functional

$$E = \int_{\mathbb{R}^2} \frac{J}{2} |\nabla \mathbf{m}|^2 + \mathcal{D}\!:\!\mathcal{L}(\mathbf{m}) + B(1 - m_z) + K_\perp\left(m_z^2 - 1\right) \mathrm{d}\mathbf{r}. \tag{5}$$

with an arbitrary (non-vanishing) spiralization tensor $\mathcal{D} \in \mathbb{R}^{2\times 2}$ emerging from Eq. (1). Here, $\mathcal{D}\!:\!\mathcal{L}(\mathbf{m})$ denotes the contraction with the chirality tensor $\mathcal{L}(\mathbf{m}) = \nabla \mathbf{m} \times \mathbf{m}$ over $x$, $y$ indices in real and spin space (see Supplementary Note 1 for details). The analysis provides a precise criterion for the optimality of antiskyrmions versus skyrmions and reveals the diversity of possible chiral phenomena including coexistence of both entities in chiral magnets. We show in this article by means of vector-spin density functional theory (DFT) calculations that the electronic structure of a double layer Fe on W(110) leads to long-range and microscopically anisotropic DM vectors that add up to tensorial elements of $\mathcal{D}$ such that the antiskyrmion becomes the non-trivial low-energy magnetization. We confirm the stability of antiskyrmions by energy minimization of Eq. (1) on a mesoscopic scale employing atomistic spin dynamics.

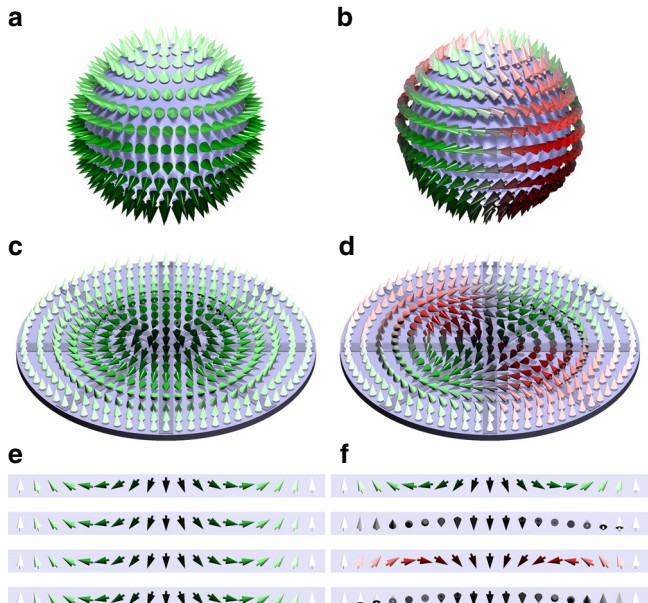

**Fig. 1** Comparison of skyrmion and antiskyrmion. **a**, **b** Néel-like skyrmion and antiskyrmion schematically shown in **c** and **d** mapped onto a sphere. The color code represents the out-of-plane component of the spins via the brightness, with *bright* (*dark*) spins pointing *up* (*down*), and their rotational sense in radial direction going from inside out changing from *red* (clockwise) via *gray* (vanishing rotational sense) to *green* (counter-clockwise). **e**, **f** Cross sections of the spin textures along the four highlighted directions shown in **c** and **d**, see also Supplementary Movie 1

## Results

**Symmetry analysis.** What may be surprising is that interface-stabilized skyrmions have exclusively been explored for (111) oriented interfaces exhibiting a $C_{3v}$ symmetry with the exception of Mn/W(100), which has a $C_{4v}$ symmetry[12]. Such high-symmetry interfaces may only support Néel- or hedgehog-type monochiral skyrmions of positive $\mathbb{S}^1$ winding number, $\nu = 1$, with the magnetization field $\mathbf{m}(\mathbf{r}) = m_\rho(\rho)\widehat{\mathbf{e}}_\rho + m_z(\rho)\widehat{\mathbf{e}}_z$ when expressed in a cylindrical coordinate system $(\rho, \varphi, z)$ and micromagnetic theory is applied. It is characterized by a fixed sense of cycloidal rotation of magnetic moments from the core outwards independent of the radial direction $\widehat{\mathbf{e}}_\rho$. The sense of rotation can be either clock- or counter-clockwise consistent with the chiral symmetry breaking. It becomes a monopole hedgehog when mapped onto a sphere, see Fig. 1a, c, e.

More general, the sense of rotation or handedness, respectively, of the magnetic structure relates the rotation of the spin **S** about the chirality vector $\mathbf{c}_\chi(\widehat{\mathbf{R}}_{ij}) = \mathbf{S}_i \times \mathbf{S}_j$ to the displacement of the spin along the direction $\widehat{\mathbf{R}}_{ij} = (\mathbf{R}_j - \mathbf{R}_i)/|\mathbf{R}_j - \mathbf{R}_i|$ of the bond between the atoms $i$ and $j$. The handedness is imposed by the DM vector $\mathbf{D}_{ij}$. According to our sign convention in Eq. (1), the energy is minimized if $\mathbf{c}_\chi(\widehat{\mathbf{R}}_{ij}) \parallel \mathbf{D}_{ij}$. We speak of right- (left-) handed or clock- (counterclock-) wise Bloch-type skyrmions if $\mathbf{c}_\chi(\widehat{\mathbf{R}}_{ij}) \cdot \widehat{\mathbf{R}}_{ij} > 0(<0)$, and of Néel-type skyrmions if $\mathbf{c}_\chi(\widehat{\mathbf{R}}_{ij}) \cdot \widehat{\mathbf{R}}_{ij}^\perp > 0(<0)$, where $\widehat{\mathbf{R}}^\perp$ is an arbitrary vector of the positive quadrant of the orthogonal complement of $\widehat{\mathbf{R}}$, e.g., formed by the vectors $\widehat{\mathbf{e}}_z \times \widehat{\mathbf{R}}$ and $\widehat{\mathbf{e}}_z$.

Monochiral skyrmions are formed when the direction of the DM vector relative to the bond direction does not change significantly for a pair of coupled spins in different crystallographic directions. Examples are systems with $C_{3v}$ or $C_{4v}$

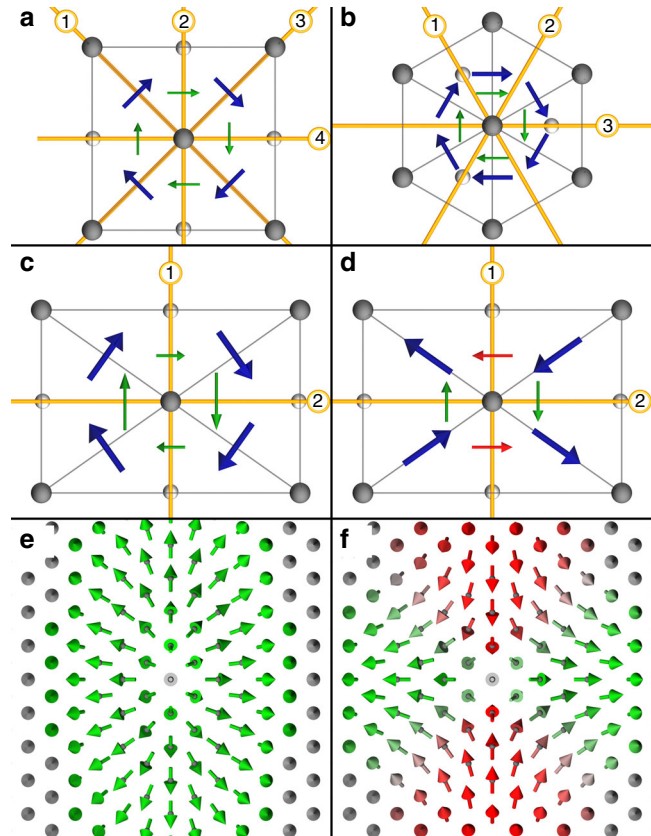

**Fig. 2** Relation between symmetry, DM vector and magnetization field. **a**, **b** Sketch of a square (hexagonal) lattice with $C_{4v}$ ($C_{3v}$) symmetry. Mirror planes (numbered from 1 to n) are indicated by *orange lines*, atoms in the surface (second) layer by *gray* (*white*) balls. The microscopic DM vectors (*blue arrows*) between n.n. sites projected onto the crystal surface are in both cases restricted by symmetry to point perpendicular to the bond. The *green arrows* represent the effective micromagnetic DM vector, $\mathbf{D}_\chi$, enforcing counter-clockwise rotating spin vectors, $\mathcal{C}_N < 0$, in all directions. **c**, **d** Two particularly chosen examples of DM vectors that match the symmetry rules of Moriya for a system with $C_{2v}$ symmetry. As discussed in the text and Supplementary Note 4, in principle the two pairs of opposing DMI vectors can point in any in-plane direction. **c**, as in **a**, **b**, DM vector to bond relation is the same in all directions. In **d** multichirality occurs, indicated by the two different colors of the effective DM vectors, which result in winding magnetic structures of opposite Néel-type chiralities, $\mathcal{C}_N \gtrless 0$. **e**, **f** Skyrmion and antiskyrmion, with magnetization textures exhibiting the shown preferred rotational senses along the different crystallographic directions. As in Fig. 1, the color code indicates the different rotational senses along the radial direction from inside out

symmetry, which appear typically at (111) or (001) oriented interfaces and surfaces of bcc and fcc crystals, respectively. This goes all the way back to symmetry arguments for DM vectors, also known as Moriya rules[4]. When applied to these high-symmetry surfaces, they determine uniquely the in-plane components of the direction of the DM vectors, $\widehat{\mathbf{e}}_{\parallel DM}$, to point perpendicular to the bond connecting the two interacting spins, see in-plane components of the nearest neighbor (n.n.) DM vectors (*blue arrow*) $\widehat{\mathbf{e}}_{\parallel DM} \parallel -\widehat{\mathbf{e}}_\varphi$ in Fig. 2a, b. Only one scalar parameter $D$ controlling the sign and absolute value along $\widehat{\mathbf{e}}_{\parallel DM}$ may vary in dependence on the electronic structure. Furthermore, the mirror symmetries of these surfaces force the DM vectors to have the same $\varphi$-component $D\varphi$ for all symmetry equivalent atom pairs. Thus, variations of the magnetization along a radial

path of any direction $\widehat{\mathbf{e}}_\rho$ will have the same preferred axis of rotation, $\mathbf{c}_\chi \parallel -\widehat{\mathbf{e}}_\varphi$ (highlighted by the *smaller green arrows*) when going along different crystallographic directions and therefore only monochiral skyrmions can be formed by the DM interaction.

The antiskyrmion (Fig. 1b, d, f), on the other hand, is the simplest form of a multichiral skyrmion[22], which is described by a polar angle $\varphi$-dependent magnetization field $\mathbf{m}(\mathbf{r}) = m_\rho(\rho,\varphi)\widehat{\mathbf{e}}_\rho + m_\varphi(\rho,\varphi)\widehat{\mathbf{e}}_\varphi + m_z(\rho)\widehat{\mathbf{e}}_z$, with negative $\mathbb{S}^1$ winding number, $\nu = -1$, characterized by different rotational senses along different radial directions away from the core (cf. Fig. 1f), thus showing multiple chiralities. It can be understood naively as an addition of a quadrupolar field to the monopole of the skyrmion (Supplementary Note 2; Supplementary Movie 2). Thus, the physical magnetization space has an orientational energy dependence relative to the lattice, a property which may add significance to the role of antiskyrmion applications. Due to the multichirality of the magnetic texture, the winding of $m_\rho$ along certain directions $\widehat{\mathbf{e}}_\varphi$ costs energy by the DMI and makes the antiskyrmion seem unfavorable over the skyrmion.

To form such multichiral structures stabilized by DMI, the DMI evidently needs to show a strong directional dependence. In films an anisotropic DMI is allowed by symmetries lower than $C_{3v}$. This is schematically sketched in Fig. 2c, d for the example of a surface with a centered-rectangular lattice exhibiting $C_{2v}$ symmetry. In this case, the number of symmetry operations is so low that the in-plane direction of the DM vectors, $\widehat{\mathbf{e}}_{\parallel \text{DM}}$, is not anymore uniquely determined by symmetry. Here, the remaining two mirror symmetries constrain only one component of the DM vector, namely the $D_z$ component to vanish, but the DM vectors are free to point into any in-plane direction.

We recall that distance vectors, $\mathbf{R}_{ij}$, between pairs of atoms $(i,j)$ form shells of symmetry equivalent vectors of equal length generated by the symmetry operations $\{\mathcal{R}\}$ mapping atom $j$ onto atom $j'$ and consequently the pseudovector $\mathbf{D}_{ij'} = \det(\mathcal{R})\mathcal{R} \cdot \mathbf{D}_{ij}$ (see, e.g., *blue arrows* in Fig. 2d, which all point along the bond direction).

**Magnetostatic energy of skyrmions and antiskyrmions.** As a consequence of the different magnetization densities, skyrmions and antiskyrmions embedded in a ferromagnetic background experience a different magnetostatic self-energy $E_{\text{mag}}$. For an infinite 2D film, $E_{\text{mag}}$ (averaged by the film thickness) decomposes to leading orders into a shape anisotropy energy, $E_{\text{mag}}^\perp$, and a magnetic film charge term $E_{\text{mag}}^\parallel$[23]. The first term depends solely on $m_z^2$, is independent of the index $\nu$ and accounts for the ferromagnetic dipolar interaction energy included already in the ferromagnetic anisotropy constant $K_\perp$. The second term is proportional to the film thickness $t$, depends on the magnetic film charge $\sigma = (\nabla \cdot m_\parallel)$ and is sensitive to the index $\nu$ and the type of skyrmions characterized by the chirality vector $\mathbf{c}_\chi(\widehat{\mathbf{R}}_{ij})$, i.e., $E_{\text{mag}}^\parallel$ is a function of $\nu$ and $\mathbf{c}_\chi$. In the axisymmetric case $E_{\text{mag}}^\parallel$ is maximal for Néel-type skyrmions, zero for Bloch-type skyrmions and precisely in the middle for arbitrary antiskyrmions, see Supplementary Note 3.

**Micromagnetic arguments for skyrmion formation.** The additional degree-of-freedom for the direction $\widehat{\mathbf{e}}_{\parallel \text{DM}}$ may allow for the formation of antiskyrmions over monochiral skyrmions. For a system with $C_{2v}$ symmetry we will verify this by inspecting in the following the winding of the magnetic structure along $\widehat{\mathbf{e}}_\rho$ from inside out in dependence on assuming only n.n. interaction for two extreme cases of DM vector directions: In the first case, the n. n. DM vectors point perpendicular to the bond as shown in

Fig. 2c (*blue arrows*). The spiralization tensor for 2D systems[24] takes then the form

$$\mathcal{D} = \frac{1}{A_\Omega}\sum_{j\in(\text{n.n.})} \mathbf{D}_{0j}\otimes\mathbf{R}_{0j} = \frac{4\mathcal{D}}{R}\begin{pmatrix} 0 & R_y^2 \\ -R_x^2 & 0 \end{pmatrix}, \qquad (6)$$

where $\mathbf{R}_{0j} = (R_x, R_y)$ is the distance vector to the n.n. atom, $R = |\mathbf{R}_{0j}|$, $\mathcal{D} = |\mathbf{D}_{0j}|$ and $A_\Omega$ is the area per surface atom.

The effective DM vector in the micromagnetic sense, along the direction of the radial coordinate $\widehat{\mathbf{e}}_\rho = (\cos(\varphi), \sin(\varphi))^\mathsf{T}$ is given by $\mathbf{D}_\chi = A_\Omega \mathcal{D}\widehat{\mathbf{e}}_\rho$ and indicated in Fig. 2 by *thin arrows* for four directions of $\widehat{\mathbf{e}}_\rho$ along the four crystallographic $\pm x$- and $\pm y$-axes, respectively. Their *green color* indicates that they result all in the same Néel-type radial chirality, $\mathcal{C}_\text{N}$, which we define as projection of $\mathbf{D}_\chi$ onto $\widehat{\mathbf{e}}_\varphi$, $\mathcal{C}_\text{N}(\varphi) = (A_\Omega\mathcal{D}\widehat{\mathbf{e}}_\rho)_\varphi$. For the case sketched in Fig. 2c it is always negative, $\mathcal{C}_\text{N} = -\frac{4\mathcal{D}}{R}(R_x^2\cos^2\varphi + R_y^2\sin^2\varphi)$, irrespective of the radial direction $\widehat{\mathbf{e}}_\rho$, i.e., the *green arrows* of Fig. 2c show in the $-\widehat{\mathbf{e}}_\varphi$ direction. The independence of $\mathcal{C}_\text{N}$ on the radial direction is analogous to interfaces with $C_{3v}$ and $C_{4v}$ symmetry and stabilizes monochiral skyrmions only (Fig. 2e). Since the micromagnetic energy function is minimized if the radial vector chirality of the spin structure $\mathbf{c}_\chi(\widehat{\mathbf{e}}_\rho)$ points parallel to $\mathbf{D}_\chi$, the negative radial chirality $\mathcal{C}_\text{N}$ corresponds to a left- or counter-clockwise magnetic structure along direction $\widehat{\mathbf{e}}_\rho$ as shown in Fig. 2e.

On the contrary, if the n.n. DM vectors point parallel to the bonds as in Fig. 2d, the off-diagonal entries of Eq. (6) change to $-4\mathcal{D}R_xR_y/RA_\Omega$ and the Néel-type chirality $\mathcal{C}_\text{N}(\varphi) = \frac{4\mathcal{D}R_xR_y}{R}(\sin^2\varphi - \cos^2\varphi)$ takes positive and negative values, depending on the radial or crystallographic direction of $\widehat{\mathbf{e}}_\rho$, as indicated by *red* and *green color*, respectively. It is now evident that multichiral antiskyrmions are stabilized by such a DM-field.

We stress that in general all directions of the DM vector are possible. They are finally system dependent and determined by the details of the electronic structure. Even when the DM vectors deviate considerably from the direction of the bond, multichiral spin textures remain lower in energy than monochiral ones as we show in the Supplementary Note 4.

A whole class of systems which possess such a $C_{2v}$ symmetry are magnetic films on the (110) surfaces of bcc or fcc crystals. Selecting a heavy metal substrate may result in a strong DM interaction[25]. In this context the prototype experimental system is certainly an Fe double layer on W(110)[26–29]. It was shown that the ground state of the system is a ferromagnet with an out-of-plane easy axis, which is favorable for skyrmion stabilization[30] and it shows a DMI, not strong enough to establish a chiral ground state, but strong enough to stabilize Néel-type domain walls of unique rotational sense[28, 29, 31]. Interestingly, a sign change in the micromagnetic DMI along different high-symmetry directions was predicted theoretically for one and two layers of Fe on W(110)[31].

**Micromagnetic selection criterion.** Relating a low-symmetry interface to a 2D spiralization tensor we could provide arguments that make the stabilization of antiskyrmions plausible. This challenges the arguments of ref. [17], which identify skyrmions as local energy minimizer within the non-trivial topological sector $Q = -1$, i.e., with $\mathbb{S}^1$ winding number $\nu = 1$. In Supplementary Note 1 we generalized the arguments to arbitrary 2D spiralization tensors $\mathcal{D}$. Our symmetry analysis exploits independent transformations $\mathcal{S}, \mathcal{R} \in O(2)$ in spin and real space, and a singular value decomposition $\mathcal{D} = (\det\mathcal{S})\mathcal{S}\mathcal{D}\mathcal{R}$ to identify a canonical form $\widetilde{\mathcal{D}} = \text{diag}(D_1, D_2)$ of thin-film DMI, accompanied by a transformation of the topological charge by a factor

$\det(\mathcal{SR}) = \pm 1$. Central results are the following:

$$\det \mathcal{D} \begin{cases} <0 & \text{implies preference of antiskyrmions} \\ >0 & \text{implies preference of skyrmions,} \end{cases} \quad (7)$$

see Theorem 1 and 2 in Supplementary Note 1. In the anisotropic case, where the $\mathcal{D}$ tensor possesses different singular values $0 < D_1 < D_2$, skyrmions and antiskyrmions coexist, see Theorem 3 in Supplementary Note 1, with approximately equal energies if $D_1 \approx 0$, i.e., $\det \mathcal{D} \approx 0$, marking a transition from a skyrmionic to an antiskyrmionic phase.

**First-principles calculations.** So far, most DM parameters calculated from first principles were either limited to n.n. inter-actions[32, 33] or to the spiralization[9, 24, 30, 31, 34], the DM strength in the micromagnetic approach, an approximation where most information about the underlying lattice structure is averaged out. Here, we go beyond these limitations and move to a more realistic description, namely the atomistic spin-lattice model (1). It includes also the pair-wise DM interactions beyond the n.n. approximation, which we directly calculate by means of DFT (see Methods section) both for atoms within the surface and interface layer but also in-between both representing the intra- and interlayer spin–spin interactions, respectively. Those pair-wise DM vectors together with the exchange constants and the on-site anisotropy are then used in spin dynamics simulations to investigate the stability of skyrmions and antiskyrmions in the system.

Our DFT results for the exchange constants $J_{ij}$ witness a strong ferromagnetic behavior both within each layer separately (n.n. values $J_{01}^{S} = 9.07$ meV and $J_{01}^{I} = 14.24$ meV for the surface and interface layer, respectively) and especially between both layers ($J_{01}^{SI} = 42.19$ meV). Going beyond the n.n. approximation allows for a far more accurate description of the physical system (the complete set of values is diagramed in Supplementary Fig. 4, Supplementary Note 5).

The calculated intralayer DM vectors are depicted in Fig. 3a, b. In the surface layer the first n.n. DM interaction dominates over all other components, while in the interface layer also the DM vectors of the second n.n. are of significant size, but of opposite direction. This oscillatory behavior can also be seen in Fig. 3d, where the distance dependence of $\text{sign}(\mathbf{D}_{ij})_\varphi |(\mathbf{D}_{ij})_\parallel|$, i.e., with respect to $|\mathbf{R}_i - \mathbf{R}_j|$ is diagramed. $(\mathbf{D}_{ij})_\parallel$ is the component of the DM vector parallel to the surface and $\text{sign}(\mathbf{D}_{ij})_\varphi$ contributing to the preferred Néel-type handedness of the spin texture ($\text{sign}(\mathbf{D}_{ij})_\varphi \rightarrow \text{sign}\,\mathcal{C}_N$). In Fig. 3c the sum of both intralayer DM interactions is superimposed on the surface plane. One can see that the contributions almost compensate each other (see e.g., the first n.n.s.), and that this compensation leads to a reduction of the absolute value of the effective n.n. interaction to the point that the next shells become significant. This shows again the importance to go beyond a n.n. approximation to describe this system.

Figure 3e displays the highly anisotropic nature of the DM vectors of this system expressed in terms of wavevectors in reciprocal space, $\mathbf{D}(\mathbf{q})$, and plotted for an area around the $\bar{\Gamma}$-point (ferromagnetic state) of the 2D Brillouin zone, an appropriate description for films with crystal periodicity. The slope of $(\mathbf{D}(\mathbf{q}))_\varphi$ close to the $\bar{\Gamma}$-point along the different crystallographic directions provides the off-diagonal elements of the spiralization tensor. For 2Fe/W(110) we obtain $\mathcal{D}_{12} = -7.85$ meV nm$^{-1}$ ($-1.26$ pJ m$^{-1}$) and $\mathcal{D}_{21} = -6.20$ meV nm$^{-1}$ ($-0.99$ pJ m$^{-1}$), respectively, which is in good agreement with the calculated values in ref. [31]. The respective values in terms of volume quantities, frequently used in micromagnetic simulations, are $-3.66$ and $-2.88$ mJ m$^{-2}$, respectively, assuming a thickness of $2d_{Fe-Fe}$ with $d_{Fe-Fe} = 0.172$ nm

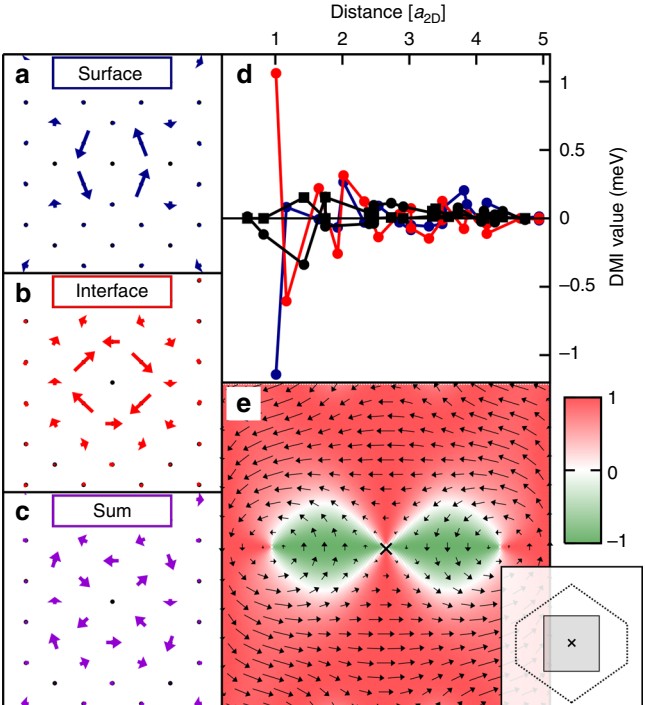

**Fig. 3** DM vectors in real and reciprocal space for 2Fe/W(110). DM vectors $\mathbf{D}_{0j}$ between atoms at site 0 and $j$ in **a** the surface, **b** the interface Fe layer and **c** the sum of both superimposed at the surface atoms in 2Fe/W(110), with atom site 0 at the center of each panel. DM vectors $\mathbf{D}_{0j}$ are placed at atom sites $j$ for visual convenience. **d** $\text{sign}(\mathbf{D}_{ij})_\varphi |(\mathbf{D}_{ij})_\parallel|$, with $(\mathbf{D}_{ij})_\parallel$ being the in-plane component of the DM vector, for the intralayer DM interaction in the surface Fe layer (*blue*), the interface Fe layer (*red*) and the interlayer DM interaction (*black circles*) for the first 21 shells of neighbors. In addition, the out-of-plane component of the interlayer DM interaction is shown (*black squares*). **e** $\mathbf{D}(\mathbf{q})$ for $\mathbf{q}$-vectors close to the center of the two-dimensional Brillouin zone (*gray marked area* in inset) for a homogeneous spin-spiral in both layers. The color code represents $(\mathbf{D}(\mathbf{q}))_\varphi / |(\mathbf{D}(\mathbf{q}))_\parallel|$

being the layer distance between the two Fe layers. The diagonal elements $\mathcal{D}_{11} = \mathcal{D}_{22} = 0$ are zero due to $C_{2v}$ symmetry and the singular values of $\mathcal{D}$ are $D_1 = |\mathcal{D}_{21}|$ and $D_2 = |\mathcal{D}_{12}|$, respectively. The determinant $\det \mathcal{D} = -\mathcal{D}_{12}\mathcal{D}_{21} < 0$, i.e., the micromagnetic antiskyrmion condition (7), which predicts antiskyrmions as the non-trivial topological magnetization soliton of lowest energy, is satisfied. In Fig. 3e, the different preferred rotation axes, $\mathbf{c}_\chi(\mathbf{q}) \propto \mathbf{D}(\mathbf{q})$, along different crystallographic directions can be clearly seen. However, it also becomes obvious that the preferred rotation axis changes for different periods of spin-spirals even if they propagate along the same direction. With this observation micromagnetic theory might come at its limit. Thus, despite the fact that the micromagnetic criterion for antiskyrmion stability is satisfied, it is advisable to analyze beyond the micromagnetic theory, whether these $\mathbf{D}$ vectors can lead to the formation of stable antiskyrmions.

We turn briefly to the MAE, $K_\perp$ in Eq. (1), exhibiting a total out-of-plane easy axis with $K_\perp = -0.11$ meV/Fe-atom. In addition to the magnetocrystalline anisotropy ($-0.25$ meV), the dipole–dipole interaction is included here as an effective on-site anisotropy (0.14 meV).

**Atomistic spin dynamics simulations.** To analyze the stability of skyrmions and antiskyrmions in 2Fe/W(110) on a mesoscopic scale, we performed atomistic spin dynamics simulations at zero

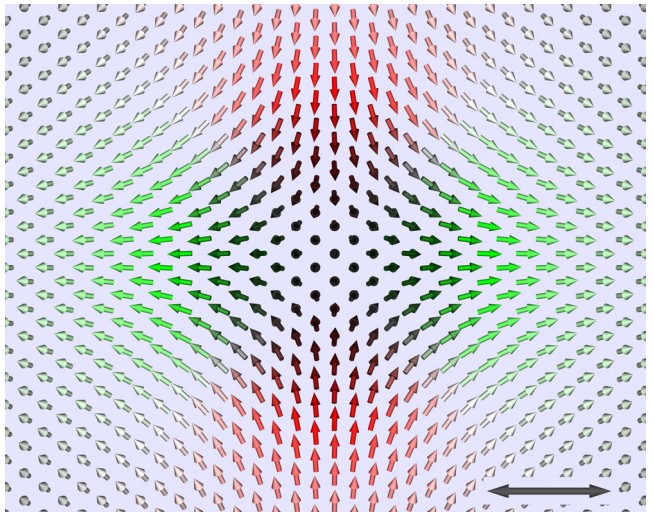

**Fig. 4** Single stable antiskyrmion in 2Fe/W(110) at zero Zeeman field. Each *arrow* represents the direction of the magnetic moment of one single atom. The color code is as in Fig. 1. For better visibility, only one-quarter of the Fe surface layer and no subsurface layer atoms are shown. The *scale bar* represents 3 nm. The antiskyrmions resulted from a simulation including interaction pairs of 8 (9) shells of neighbors for the intralayer (interlayer) coupling equivalent to a total of 26 (26) neighbors

temperature using the extended first-principles Heisenberg model (1) with parameters determined above (see Methods section).

While our simulations confirm the ferromagnetic state to be the ground state, antiskyrmions (see Fig. 4 for a top view of the resulting structure) appear as stable states for the interaction parameters describing 2Fe/W(110). They can be found starting from different initial configurations, e.g., from antiskyrmions with a broad range of diameters, irrespective of their spatial orientation, as well as from random configurations. We also investigated the stability of the antiskyrmion with respect to the different number of shells of neighbors resolved in the first-principles calculations and subsequently taken into account in the spin dynamics simulations. Stability of antiskyrmions was found for all tested numbers of shells, however, the exact diameter of the antiskyrmion might vary. Furthermore, for some numbers of shells, a small magnetic field was required to stabilize the antiskyrmion, while for most cases, the antiskyrmion was stable even at zero magnetic field. However, in all tested cases the required field was lower than 150 mT, equivalent to about 0.02 meV/Fe-atom, which lies clearly within the limit of first-principles calculations.

The anisotropy of the micromagnetic $\mathcal{D}$-tensor, $2|D_1 - D_2|/(D_1 + D_2)$ amounts to 23%. Although the anisotropy is small, our micromagnetic theory suggests a possible coexistence of skyrmions with antiskyrmions. Starting the atomistic spin dynamics simulations from any chiral skyrmion, we observe their collapse into the ferromagnetic state if no external magnetic field is applied. Applying an magnetic field parallel to the core of the skyrmion, we observe in our simulations that at about 700 mT (the exact value depends again on the number of shells which are taken into account) the skyrmion no longer collapses. Instead, the skyrmion expands toward the boundary of our simulated lattice resulting in a ferromagnetic state aligned parallel to the magnetic field. Fine tracking the stability of skyrmions with respect to the external magnetic field we can say that within a field resolution of 1 mT no stable skyrmion was found. A further analysis of the role of different DM energy contributions (intra- and interlayer coupling) on the stabilization of antiskyrmions can be found in the Supplementary Note 6.

**Discussion**

The interplay of energy and topology resulting in the optimality of certain topological states in micromagnetics is revealed by general principles of mathematical analysis. In the context of 2D systems where DMI is represented by a $2 \times 2$ spiralization tensor $\mathcal{D}$, it follows that the energetically selected homotopy class (skyrmion versus antiskyrmion) is determined by the orientation of $\mathcal{D}$, i.e., the sign of $\det \mathcal{D}$ (see Theorem 1 in Supplementary Note 1). Chiral features of the skyrmion texture are determined by the tensorial components of $\mathcal{D}$. For example, right- (left-) handed Néel-type skyrmions typical for interfaces are obtained if sign $(\mathcal{D}_{21} - \mathcal{D}_{12}) = 1$ $(-1)$ provided $\mathcal{D}_{21} \neq \mathcal{D}_{12}$.

The prototypical forms of DMI arising in ultra-thin films or from 2D reductions of the cubic B20 type satisfy $\det \mathcal{D} = D^2 > 0$ where $D \neq 0$ is the DMI constant. As is known[17], chiral skyrmions with $Q = -1$ occur as local energy minimizers within the field polarized regime $BJ \gtrsim D^2$. The crucial observation is that the least energies $E_Q$ in the topological sectors of charge $Q = \pm 1$ are separated: $E_{-1} < 4\pi J$ while $E_1 = 4\pi J$, where the energy quantum $4\pi J$ is the threshold for the collapse of a topological entity. Consequently, antiskyrmionic configurations may release energy at the expense of forming a point singularity.

Tuning the system toward $\mathcal{D}$ with negative determinant, as in the example of an Fe double layer on W(110), lowers the least energy in the topological sector $Q = 1$ below $Q = -1$, i.e., preference of antiskyrmions. A particularly interesting situation arises in the limiting case of vanishing determinant, i.e., the presence of effectively only one DM vector, where our analysis predicts the coexistence of skyrmions and antiskyrmions with equal energies. Coexistence, however, is not exclusive to this extreme situation. Whenever the singular values $D_1$ and $D_2$ of $\mathcal{D}$ differ, the least energies in both topological sectors $Q = \pm 1$ are subcritical $E_Q < 4\pi J$. This allows for the occurrence of both entities, while $\det \mathcal{D}$ serves as a measure of the difference of their energies.

The magnetostatic energy is not able to change the handedness of the skyrmion, but can be used to tune the energy difference between the skyrmion and antiskyrmion on a fine energy scale as function of the film thickness.

Since the $\mathcal{D}$ tensor is a measurable as well as computable quantity, the sign and value of the determinant serves as a simple but powerful criterion for the occurrence and stability of skyrmions and antiskyrmions as well as an effective property descriptor to navigate the search for materials. It serves also as classification scheme of chiral magnets into isotropic rank-three DMI bulk and rank-two DMI film magnets, with a DMI described by a single constant, the spiralization, for which antiskyrmions are stable only for bulk crystals with certain point group symmetries. Then, we have the anisotropic rank-two DMI film magnets, where skyrmions and antiskyrmions can coexist, while the sign of $\det \mathcal{D}$ determines which of the two has the lower energy. Finally, zero determinant indicates a rank-one DMI material, for which skyrmions and antiskyrmions have the same energy.

Our theoretical predictions are consistent with the calculations of Güngördü et al.[21] observing the occurrence of skyrmion or antiskyrmion lattices depending on details of effective spin–orbit models. Their theoretical investigation highlights the role of global transformations and in particular the topology reversing effect of improper rotations applied to the spiralization tensor. Our analysis now provides a precise selection criterion in the case of general $2 \times 2$ spiralization tensors. Coexistence of isolated skyrmions and antiskyrmions for anisotropic DMI is not only an unexpected theoretical result in the field of chiral skyrmions, but may also lay the groundwork for spintronic applications.

Combining DFT with atomistic spin dynamics simulations, in this paper we showed for a double layer Fe on W(110) that the interplay of Heisenberg and anisotropic Dzyaloshinskii–Moriya exchange with uniaxial magnetocrystalline anisotropy and modest external magnetic field results in fact in the stabilization of antiskyrmions. The spatially anisotropic DMI vectors, $\mathbf{D}_{ij}$, enabled by the $C_{2v}$ symmetry sum up to a spiralization tensor $\mathcal{D}$, rather than a spiralization scalar $D$, with negative determinant consistent with the micromagnetic antiskyrmion criterion. The tensor is surprisingly isotropic as evidenced by the singular values $D_1 \approx D_2$ and no coexistence of skyrmions and antiskyrmions was found by means of the spin dynamics. Details of the electronic structure lead to pair interactions of the DMI of competing sign, showing that a n.n. approximation is insufficient to resolve the stability, shape, orientation and fine structure of the antiskyrmion. As seen from Fig. 4 the symmetry axis of the antiskyrmions matches with the underlying crystal lattice in that the counter-clockwise rotation of the magnetic moments follows the long axis of the unit cell and the clockwise the short one.

Considering that heavy transition metal substrates[30] like W contribute most to the DMI and that their complex Fermi surfaces[35] lead to rather anisotropic $D_{ij}$ parameters, together with the $C_{2v}$ symmetry, we consider 2Fe/W(110) as a prototypical example of a much wider class of magnetic films and heterostructures that host antiskyrmions rather than skyrmions as topologically non-trivial stable states. Other members of this class are those with (110) oriented interfaces between $3d$ and $5d$ transition metals. In fact, an anisotropic DMI was recently measured in a thin epitaxial Au/Co film on W(110)[36], but this system favored elliptical skyrmions instead of antiskyrmions. Other candidates are $3d$ metals on semiconductor (100) and (110) surfaces, e.g., Fe on Ge or GaAs, exhibiting $C_{2v}$ and $C_s$ symmetry, respectively.

In this article we focused on interface-stabilized antiskyrmions typical for thin magnetic films and heterostructures, but our arguments apply also to bulk systems with broken inversion symmetry and space groups, for which the effective spiralization tensor is described by a $2 \times 2$ matrix rather than a scalar. This motivates the synthesis of magnetic materials, e.g., with $C_{2v}$ symmetry on the basis of quaternary selenides such as InVSe$_2$O$_8$[37]. When typical chiral bulk magnets such as B20 alloys, with their lack of bulk inversion symmetry ($\mathbf{r} \nrightarrow -\mathbf{r}$) and a DMI described by a spiralization scalar $D$, meet the inversion asymmetry ($z \nrightarrow -z$) of surfaces, films and heterostructures, we expect a non-zero $\mathcal{D}$-tensor. Thus, antiskyrmions can emerge at interfaces of skyrmion-carrying bulk phases. To complete the discussion, antiskyrmions in films with symmetries resulting in a scalar spiralization do not exist, but can be found in bulk crystals with space groups $D_{2d}$ and $S_4$. The most promising systems to host antiskyrmions seem to be magnetic Heusler alloys (e.g., Mn$_2$RhSn[38]), chalcopyrites (e.g., CuFeS$_2$[39]), stannites (e.g., Cu$_2$FeSnSe$_4$[40]) or kesterites (e.g., Cu$_2$Mn$_{1-x}$Co$_x$SnS$_4$[41]).

Since the topological charge is related to the gyrovector and the Magnus force acting on the antiskyrmion, we expect in analogy to the skrymion Hall effect[42] an antiskyrmion Hall effect. Thus, our work prompts the exploration of the dynamical, dissipative, and transport properties related to the additional orientational stabilization of the antiskyrmions with respect to the underlying lattice in comparison to the monochiral skyrmions.

This work also motivates the design of materials that can host skyrmions and antiskyrmions simultaneously (the case $0 < D_1 < D_2$). This opens an exciting perspective to investigate the interaction of particle and antiparticle of different energies in the spirit of the skyrmion to antiskyrmion interaction in dipolar magnets[19] or frustrated magnets[43] and the conditions of mixed ordered lattices and phases. It should be explored in how far the tunneling mixing magneto-resistance[44, 45] effect can be used to discern electrically skyrmions from antiskyrmions. We expect that the topological charge density of an antiskyrmion produces an emergent magnetic field opposite to that of skyrmions for the same material. These different emergent fields give rise to topological orbital moments of opposite sign that can be exploited to discriminate the different skyrmion–antiskyrmion phases spectroscopically using soft X-ray magnetic circular dichroism[46]. These mixed phases are lattices of staggered magnetic fields. The topological Hall effect of these lattices would be an exciting topic to study.

Rank-one materials, i.e., DMI materials with $\det \mathcal{D} = 0$, should be particularly interesting for information storage and processing. One may envisage a creation process of a skyrmion–antiskyrmion pair out of the trivial FM state or a domain wall keeping the total $\mathbb{S}^2$ winding number, $Q$, zero. They allow for an extension of the skyrmion race track idea[2], where the information is encoded in the relative positions or time sequences, respectively, of the skyrmions along the track, to a skyrmion–antiskyrmion race track memory, where the binary information is encoded in the sequence of skyrmions and antiskyrmions, which are expected to be read out distinctly. Upon further investigations of skyrmion–antiskyrmion interactions in constrictions, rank-one materials may be an ideal host to extend the concepts of skyrmion logic gates[47] to magnetic logic gates in which skyrmions and antiskyrmions are the elementary particles for binary operations.

## Methods

**Mathematical analysis**. Rather than attempting to solve the equilibrium equations directly, we examine the micromagnetic energy landscape over different homotopy classes. Key quantities are least energies (5) over homotopy classes characterized by $Q \in \mathbb{Z}$, i.e.,

$$E_Q = \inf\{E(\mathbf{m}) : Q(\mathbf{m}) = Q\}, \tag{8}$$

which may not be attained in general. Chiral skyrmions and antiskyrmions arise as relative energy minimizers, which are stable with respect to arbitrary perturbations in the configuration space. The proof of existence and coexistence is based on variational arguments combining constructive upper and ansatz-free lower energy bounds. The bounds particularly capture the energetic optimality of antiskyrmionic versus skyrmionic configurations (or vice versa) leading to the micromagnetic selection criterion (7). Crucially in the context of anisotropic DMI, the energy-based approach does not rely on any symmetry assumption.

By virtue of the symmetry argument preceding (7), it is sufficient to show that skyrmions ($Q = -1$) are preferred for spiralization tensors of the canonical form $\mathcal{D} = \mathrm{diag}(D_1, D_2)$ with positive singular values $D_1$ and $D_2$. In the anisotropic case $D_2 > D_1$, suitable trial fields in form of modified stereographic maps yield the upper energy bounds

$$E_Q < 4\pi J \quad \text{for} \quad Q = \pm 1, \tag{9}$$

valid for arbitrary Zeeman fields and anisotropies. In particular, $E_{\pm 1}$ lie strictly below the threshold of collapse. Notably, in the isotropic case $D_1 = D_2$ the upper bound holds only for $Q = -1$, and antiskyrmions may collapse. Bogomolny-type arguments yield the lower bounds

$$E_Q > E_1 > E_{-1} \quad \text{for all} \quad |Q| > 1, \tag{10}$$

valid in a regime of large fields, where the energy is non-negative. In this regime skyrmionic configurations can produce lower energies than any antiskyrmionic configuration. Using Eqs. (9) and (10), existence of skyrmions and antiskyrmions can be deduced by virtue of a concentration-compactness argument as in ref. [17].

Notably, in the rank-one case $D_1 = 0$ and $D_2 > 0$, it follows from a reflection argument that $E_Q = E_{-Q}$ for all $Q \in \mathbb{Z}$. In particular, skyrmions and antiskyrmions coexist with the same energy. Details are provided in Supplementary Note 1.

**First-principles calculations**. We performed vector-spin DFT calculations using the film version of the full-potential linearized augmented plane wave method as implemented in the FLEUR code (www.flapw.de). By using the FLEUR code, we get access to the total energy of non-collinear magnetic structures and spin-spirals both with and without spin–orbit coupling, which allows to obtain the parameters entering Eq. (1). For details on the computations and choice of exchange correlation potential we refer to Supplementary Note 5.

To calculate the pairwise DM vectors from DFT, we extended the derivation of interatomic exchange interactions in ferromagnets by Ležaić et al.[48]

(which we also used here to calculate the exchange constants $J_{ij}$) to the interatomic DMI parameters $\mathbf{D}_{ij}$. These have then been evaluated by means of the FLEUR code carrying out coned spin-spirals calculations for a grid of wavevectors $\mathbf{q}$ with a small cone angle $\theta$ employing the force theorem (for details see Supplementary Note 5). The results have been cross-checked against independent Korringa–Kohn–Rostocker Green-function calculations and we obtained qualitative and quantitative agreement (Supplementary Note 7).

**Atomistic spin dynamics simulations**. In order to analyze the stability of skyrmions and antiskyrmions we have relaxed approximate and random spin configurations on a lattice with $(150 \times 150)$ spins in each layer of the Fe double layer at zero temperature according to the Landau–Lifshitz–Gilbert equation of spin dynamics:

$$\hbar \frac{d\mathbf{S}_i}{dt} = \frac{\partial H}{\partial \mathbf{S}_i} \times \mathbf{S}_i - \alpha \left( \frac{\partial H}{\partial \mathbf{S}_i} \times \mathbf{S}_i \right) \times \mathbf{S}_i \qquad (11)$$

using the extended Heisenberg model (1), where $H$ is the Hamiltonian, $\mathbf{S}_i$ is the spin of unit-length at site $i$ and $\alpha$ is the damping parameter. To achieve fast relaxation from initial distortions of the spin system along the physical path toward lower energy, we have used a range of $\alpha = 0.7$–$0.1$ and time steps ranging from 0.3 to 3 fs, respectively. We included the interaction parameters of the neighbors shown in Fig. 3 and in the Supplementary Fig. 4 (Supplementary Note 5) for the pairwise contributions including up to 21 shells with in total 71 pairs per atom for both the intralayer and the interlayer coupling. We have carried out simulations for periodic boundary conditions, so as to avoid influence of the boundary on the stabilization of the antiskyrmion. The equation of motion was integrated by applying an efficient and robust semi-implicit numerical method with built-in angular momentum conservation[49], as implemented in Spirit (spirit-code.github.io). States, which are stabilized this way in spite of the different initial perturbations or distortions, we experienced as local minima in the energy landscape described by the Hamiltonian.

**Code availability**. The DFT code FLEUR and the spin dynamics code Spirit used in this work are publicly accessible at www.flapw.de and spirit-code.github.io, respectively.

**Data availability**. The data that support the findings of this study are available from the corresponding authors upon request.

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

## Acknowledgements

We would like to thank Miriam Hinzen, Frank Freimuth, Yuriy Mokrousov, and Hannes Jónsson for fruitful discussions. We gratefully acknowledge computing time on the JURECA supercomputer provided by the Jülich Supercomputing Centre (JSC). B.Z. and S.B. acknowledge funding from the European Union's Horizon 2020 research and innovation programme under grant agreement number 665095 (FET-Open project MAGicSky). G.P.M. acknowledges funding from the Icelandic Research Fund (grant no. 152483-052). C.M. acknowledges funding from Deutsche Forschungsgemeinschaft (DFG grant no. ME 2273/3-1). C.M. and S.B. acknowledge seed-fund support from JARA-FIT.

## Author contributions

M.H. and B.Z. conceptualized and carried out the DFT calculations. G.P.M., D.S., M.H. and N.S.K. developed a spin dynamics code and M.H. and G.P.M. performed the spin dynamic simulations. C.M. provided the mathematical analysis of the micromagnetic functional. S.B. initiated this work. All authors took part in the analysis and the discussion of the results and contributed to the writing of the paper.

## Additional information

**Competing interests:** The authors declare no competing financial interests.

