## [Peer Review File · Nature Communications]

Reviewers' comments:

Reviewer #1 (Remarks to the Author):

This paper does not seem appropriate for Nature Comm., which seeks to publish works of general interest. The paper may be accessible only to a small community of people which are well familiar with the particular topic of skyrmion patterns in magnetic media. The text is quite verbose, but it is not written in a form that would be friendly to the potentially broad readership. In particular, while the central topic is the use of the Dzyaloshinskii - Moriya interaction, it is not even explicitly defined in the paper, assuming that it is known to everyone, which cannot be correct. Further, while the paper is a fully theoretical one, and its stated objective is to demonstrate that antiskyrmions may realize a physically relevant stable state, the text only briefly mentions that the analysis is carried out by means of a variational method, but no details of the analysis - which is, obviously, the core part of the work - are displayed. Instead, readers are suggested to look for "mathematical details" in some supplementary material, which is not a part of the paper. As a result, the entire text seems as an extended abstract (a lengthy one, which would, nevertheless, be quite obscure to non-specialists), rather than a systematically written article.

It may be that the prediction of the stability of the antiskyrmion, announced in the paper as its main result, is an essential prediction, that may stimulate new experiments (magnetic media, in which the authors expect a possibility to realize the prediction, are briefly mentioned in the paper). However, if the paper is intended for the publication in a broad-interest journal, such as Nature Comm., the text should be made accessible to the broad readership, and it should definitely be made self-contained, providing explicit definitions of the basic concepts in the narrow topic under the consideration, rather than assuming that they are known to everyone (which may sometimes be acceptable in papers submitted to a specialized journal).

It may be recommended to rewrite the paper in the self-consistent form, including the core proof of the antiskyrmion's stability, rather than concealing it in supplementary files, and resubmit the work to one of leading solid-state-physics journals, such as Phys. Rev. B or J. Phys. Cond. Matt.

Reviewer #2 (Remarks to the Author):

This paper shows that anisotropic Dzyaloshinskii-Moriya interaction at a low-symmetry surface can stabilize antiskyrmions relative to skyrmions. A model is presented and described in several different ways (analytical theory, micromagnetic and atomistic simulations), and a specific system (Fe bilayer on W(110)) is predicted using first-principles calculations. The message is clear and timely, and I recommend this paper for publication. The authors may want to cite the experimental paper on arXiv (which has appeared after the present paper) reporting the observation of antiskyrmions: <https://arxiv.org/abs/1703.01017>.

I have a few comments that the authors may consider before publication:

1. I don't understand why the D_z component should vanish (line 232) in Fig. 2 at the surface with C_{2v} symmetry. Why can't it alternate (up and down) among the four nearest neighbors?
2. I find Fig. 2 (c,d) somewhat misleading without an in-depth examination. These two orientations of the DM vectors are not fixed by any symmetry; there is no reason for them to be perpendicular or parallel to the bonds in any real system. I suggest adding a panel with a generic orientation and make it clear that panels (c) and (d) show directions that are chosen somewhat arbitrarily.
3. Lines 342-344: different exchange parameters should not carry the same label (J_{01}).

4. Line 486: it is unclear what is meant by the "energy discrepancy."

5. Typo in the supplementary: "slap" should be "slab"

Reviewer #3 (Remarks to the Author):

Reviewer Comments on NCOMMS-17-01580A-Z "Antiskyrmions stabilized at interfaces by anisotropic Dzyaloshinskii-Moriya interaction"

Remarks to the Author

This manuscript presents a theoretical proposal of antiskyrmions stabilized by anisotropic Dzyaloshinskii-Moriya interaction.

Skyrmion has been a very hot topic in the field of condensed matters since its first experimental observation in magnetic materials in 2009. This manuscript deals with the antiskyrmions which have been explored in this literature [e.g., Nature Commun. 7, 10542 (2016) etc.]. Nevertheless, Ref. [19] pointed out that antiskyrmions are unstable in a more symmetric system, whereas the present manuscript have carried out systematic theoretical and numerical studies of antiskyrmion stabilities in chiral magnets. This might be an important research advance of antiskyrmions since it has some novel properties as compared to the conventional skyrmions which have been intensively studied in the literature. For example, it may add a new dimension and new degree of freedom in skyrmionic devices, such as manipulating its helicity etc for spin logic applications [see, Scientific Reports 5, 9400 (2015)]. In a recent arxiv paper on frustrated skyrmion [arXiv:1703.07501], it is shown that frustrated antiskyrmions is stable solution and the antiskyrmion can interact with skyrmion in a fascinating way. So it would be interesting to compare this manuscript with the arxiv paper briefly. In addition, I hope the authors can provide a brief discussion on the practical application of antiskyrmions, which might be beneficial for a wide readership.

In my view, the main results of interface stabilized antiskyrmions will contribute to the skyrmionic research by adding new dimension to this exciting field. It presents something very important and will motivate further research efforts towards its experimental realization. Therefore I support the publication of this paper in Nature Communications provided the above comments can be addressed.

Response to Referees

We would like to thank all reviewers for the careful reading of our manuscript and for the many insightful comments and observations. In the following we present our detailed replies to all the points that have been raised.

Please notice the blue color indicates changes of the manuscript.

To all reviewers:

- (1) Briefly after the submission of our paper we got aware of some papers uploaded to arXiv after our paper was submitted. Some of those have already been identified by you and we include them and comment on those case by case at the individual comments to the reviewers. Among them was a paper on Co/W(110) (<https://arxiv.org/abs/1701.05062>). This has been added now to our paper: We included the following sentence in the Discussion section:
Other members of this class are certainly those with (110) oriented interfaces between 3d and 5d transition-metals. In fact, an anisotropic DMI was recently measured in a thin epitaxial Au/Co film on W(110) \cite{Camosi:arXiv}, but this system favored elliptical skyrmions instead of anti-skyrmions. Other candidates are 3d metals on semiconductor (100) and (110) surfaces, e.g., Fe on Ge or GaAs, exhibiting C2v and Cs symmetry, respectively.
- (2) During the March Meeting of the American Physical Society and the German Physical Society (the largest March meeting in Europe) we noticed that the topic of antiskyrmions comes up, but there is much confusion in the field of antiskyrmions. To clarify this, we introduce a proper distinction and classification of different antiskyrmion carrying material systems: the isotropic rank-three- and rank-two-DMI materials, the anisotropic rank-two-DMI materials and the rank-one materials systems that have antiskyrmions of different properties. We introduce in the Discussion section: It serves also as classification scheme of chiral magnets into isotropic rank-three-DMI bulk and rank-two-DMI film magnets, with a DMI described by a single constant, the spiralization, for which antiskyrmions are stable only for bulk crystals with certain point group symmetries. Then, we have the anisotropic rank-two-DMI film magnets, where skyrmions and antiskyrmions can coexist, and the sign of \$\det \mathbf{D}\$ determines which of the two has the lower energy. Zero determinant determines then rank-one-DMI material, for which skyrmions and antiskyrmions have the same energy.
- (3) In the third-last paragraph of the section before the start of the section: Results, we introduced the terms Bloch-type and Néel type skyrmions clearer: We speak of right- (left-) handed Bloch-type skyrmions if \$\mathbf{v}_c \cdot \chi(\hat{\mathbf{v}}_R)_i > 0\$, (\$< 0\$ ), and of Néel-type skyrmions if \$\mathbf{v}_c \cdot \chi(\hat{\mathbf{v}}_R)_i \cdot \hat{\mathbf{v}}_R^\perp > 0\$, (\$< 0\$ ), where \$\hat{\mathbf{v}}_R^\perp\$ is an arbitrary vector of the positive quadrant of the orthogonal complement of \$\hat{\mathbf{v}}_R\$, \$\mathbf{e}_z\$ formed by the vectors \$\mathbf{e}_z \times \hat{\mathbf{v}}_R\$ and \$\hat{\mathbf{v}}_R \times \mathbf{e}_z\$.

(4) Re-reading our manuscript we found a few minor spelling errors that we changed without further notice in the manuscript (cannot only -> can not only; possesses -> possess; offers -> offer; whereas -> where; on -> of).

(5) In the section: Discussion, paragraph 2, we corrected a mistake in the following mathematical expression from $BJ \geq D^2$ to $BJ > D^2$.

(6) We have moved the following two sentences:

The handedness of the skyrmion texture is determined by the tensorial components of \mathcal{D} . Right- (left-) handed N\`eel-type skyrmions typical for interfaces are obtained if $\text{sign}(\left(\mathcal{D}_{21} - \mathcal{D}_{12}\right)) = 1 \ (-1)$ provided $\mathcal{D}_{21} \neq \mathcal{D}_{12}$.

from the second paragraph of the section Discussion to the first one in order to improve the flow of reading. This is not further indicated in the text.

(7) Re-reading our manuscript, we realized that it can be improved by additional words or we found a few more elegant formulations:

- ... is **typically** of minor importance ...
- changes to become -> **is replaced by**
- **The index** v is also referred to as...
- Note v does not depend on -> **Being independent of**
- skyrmions and antiskyrmions are **also** distinct in their handedness properties
- monochiral skyrmions of **positive** \mathbb{S}^1 winding number
- which, when applied to these high-symmetry surfaces,...
- ..., as for instance ... -> **Examples are** ...

Reviewer #1: This paper does not seem appropriate for Nature Comm., which seeks to publish works of general interest. The paper may be accessible only to a small community of people which are well familiar with the particular topic of skyrmion patterns in magnetic media. The text is quite verbose, but it is not written in a form that would be friendly to the potentially broad readership.

Reply: We agree with the referee that a Nature paper should address subjects as well as presentations in such a way that it is accessible to broad readership as much as possible. But in Nature Communications we should be also balance with the mission statement of this journal which states: *Papers published by Nature Communication represent important advances of significance to specialists within each field.*

We do not quite agree with the statement that the paper is only accessible to a small community. Currently the field of magnetic skyrmions is a field which experiences a strong expansion and spread of interest, in which quite different communities meet: Mathematics, magnetism, electronic properties, transport, dynamics, Terahertz radiation, materials science, devices, information technology, etc. E.g. The National Science Foundation of Germany (DFG) has just initiated a national priority program on Skyrmionics to support this research, and in America, DARPA is launching a support program under the name texitronics. We would like to underpin our statements above mentioning that at the March meeting of the American Physical Society (APS) we witnessed 2 sessions fully dedicated to Skyrmions and 34 sessions that contained talks about skyrmions, the March Meeting of the German Physical Society (the largest in Europe) had 8 sessions dedicated to skyrmions and 29 sessions

which contained talks about skyrmions, and the Web of Science search of citations (left figure) and papers published (right figure) discussing magnetic skyrmions (black) and skyrmions in all fields of science (red) shown here as function of publication year, shows clearly a rapid interest in this field. It is definitely not a small group of people, instead it is part of the topological revolution in condensed matter physics.

Reviewer #1: In particular, while the central topic is the use of the Dzyaloshinskii - Moriya interaction, it is not even explicitly defined in the paper, assuming that it is known to everyone, which cannot be correct.

Reply: Well, we are not sure we can agree on this statement. We introduce the Dzyaloshinskii-Moriya interaction (DMI) in equations (1) to (3) in terms of the spin-lattice model as well as in the micromagnetic model in simple cubic as well as the most general form. We also briefly mention its origin: *The DMI results from the spin-orbit interaction and is only non-zero for solids lacking bulk or structure inversion symmetry.* We have introduced the interface induced Dzyaloshinskii-Moriya in the following paper: Bode, Heide, von Bergmann, Heinze, Bihlmayer, Kubetzka, Pietzsch, Blügel, Wiesendanger, Nature 447, 190 (2007). Since then, 10 years have passed. We, as well as many other people have calculated the strength of parameters, have explained the quantities on the basis of analytical and micromagnetic models. Therefore, we decided not to repeat this, instead having a steep entrance into the field, focusing on the novelty relative to the past.

Reviewer #1: Further, while the paper is a fully theoretical one, and its stated objective is to demonstrate that antiskyrmions may realize a physically relevant stable state, the text only briefly mentions that the analysis is carried out by means of a variational method, but no details of the analysis - which is, obviously, the core part of the work - are displayed. Instead, readers are suggested to look for "mathematical details" in some supplementary material, which is not a part of the paper. As a result, the entire text seems as an extended abstract (a lengthy one, which would, nevertheless, be quite obscure to non-specialists), rather than a systematically written article.

Reply: The Discussion and Methods sections aim to provide a concise and widely understandable account on the mathematical ideas involved. To keep the paper well accessible to the broad readership of Nature Communications and (hence) well citable we have decided to include the mathematical details in the supplement. Everybody has access to these supplementary materials. We are very happy and proud of the mathematical analysis, but equally to the atomistic spindynamics and the first-principles

calculations. This interdisciplinary approach of mathematics, computations and simulations make-up the value of our paper and therefore we have chosen a two-level approach. At first, we present in the main text the overall introduction, the results and a discussion of those to the reader, but provide on the second level more technical information so that all readers can reproduce our results.

Reviewer #1: It may be that the prediction of the stability of the antiskyrmion, announced in the paper as its main result, is an essential prediction, that may stimulate new experiments (magnetic media, in which the authors expect a possibility to realize the prediction, are briefly mentioned in the paper). However, if the paper is intended for the publication in a broad-interest journal, such as Nature Comm., the text should be made accessible to the broad readership, and it should definitely be made self-contained, providing explicit definitions of the basic concepts in the narrow topic under the consideration, rather than assuming that they are known to everyone (which may sometimes be acceptable in papers submitted to a specialized journal).

Reply: Summarizing above arguments, we think the paper is self-contained. It provides all definitions through equations (1) ... (3) and together with the supplementary materials, which we consider an important part of the paper, the reader has all necessary information to reproduce the results.

Reviewer #1: It may be recommended to rewrite the paper in the self-consistent form, including the core proof of the antiskyrmion's stability, rather than concealing it in supplementary files, and resubmit the work to one of leading solid-state-physics journals, such as Phys. Rev. B or J. Phys. Cond. Matt.

Reply: Since we are convinced that one particular value of the paper lies in the combination of different disciplines, and since we are convinced that this multi-disciplinary approach enables the best advancement of science in this field, we refrain from chopping the paper into pieces and publish the results in parallel in different journals.

Reviewer #2: This paper shows that anisotropic Dzyaloshinskii-Moriya interaction at a low-symmetry surface can stabilize antiskyrmions relative to skyrmions. A model is presented and described in several different ways (analytical theory, micromagnetic and atomistic simulations), and a specific system (Fe bilayer on W(110)) is predicted using first-principles calculations. The message is clear and timely, and I recommend this paper for publication.

The authors may want to cite the experimental paper on arXiv (which has appeared after the present paper) reporting the observation of antiskyrmions:

<https://arxiv.org/abs/1703.01017>.

Reply: We are happy to cite the paper. The sentence reads now: *Some of which, e.g. those transforming to bulk crystals with S_4 or D_{2d} symmetry, can lead to the stabilization of antiskyrmions rather than skyrmions, as has been recently demonstrated for a acentric tetragonal MnPtPdSn Heusler compound~\cite{Nayak:arXiv}*

Reviewer #2: I have a few comments that the authors may consider before publication:

Comment 1. I don't understand why the D_z component should vanish (line 232) in Fig. 2 at the surface with C_{2v} symmetry. Why can't it alternate (up and down) among the four nearest neighbors?

Reply: The vanishing z-component of the DMI vectors is a direct result of the remaining mirror symmetries of the C_{2v} point group. There are various ways to look at that: (i) the DMI-vector is three-dimensional, having appropriate symmetry operations can nullify one or two components. (ii) Consistent with the two mirror planes, the C_{2v} point group has an additional 2-fold rotational axis normal to the surface at the bond center between two nearest neighbor atoms. Then from the fourth Moriya rule (Physical Review 120, 91 (1960)), it follows that the DM vector stands perpendicular to this axis and thus parallel to the surface. As it can be seen from Fig. 3, this rule does apply for any pairs of neighbors within one layer, however, for the interlayer coupling this symmetry rule is not anymore fulfilled and a non-vanishing z-component is possible.

We modify the following sentence slightly to make this clearer: Here, the remaining **two mirror** symmetries constrain only one component of the DM vector, namely the D_z -component to vanish, but the DM vectors are free to point into any in-plane direction.

Comment 2. I find Fig. 2 (c,d) somewhat misleading without an in-depth examination. These two orientations of the DM vectors are not fixed by any symmetry; there is no reason for them to be perpendicular or parallel to the bonds in any real system. I suggest adding a panel with a generic orientation and make it clear that panels (c) and (d) show directions that are chosen somewhat arbitrarily.

Reply: We change the sentence in the figure caption from:

Two exemplary cases of DM vectors consistent with Moriya's symmetry rules for a system with C_{2v} symmetry. To:

Two particularly chosen examples of DM vectors that match the symmetry rules of Moriya for a system with C_{2v} symmetry. As discussed in the text and Supplementary Note 3, in principle the two pairs of opposing DMI vectors can point in any in-plane direction.

Maybe the last sentence is sufficient to avoid a further panel with pairs of atoms pointing in an arbitrary direction.

Comment 3. Lines 342-344: different exchange parameters should not carry the same label (J_{01}).

Reply: We understand your argument. To keep a high level of readability, we wanted to avoid the introduction of further indices to the exchange coupling such as the layer index. Therefore, we have chosen an explicit wording of the exchange parameters like J_{01} of surface layer and interface layer. If you think it is too confusing and certainly not 100% correct, we will add the indices. Thus, we use the notations $J_{01}^S, J_{01}^I, J_{01}^{SI}$.

Comment 4. Line 486: it is unclear what is meant by the “energy discrepancy.”

Reply: We agree this is a not well-chosen expression. The sentence reads now: This allows for the occurrence of both entities, while $\det D$ serves as a measure of the difference of their energies.

Comment 5. Typo in the supplementary: “slap” should be “slab”

Reply: Thank you, we corrected this word in Supplementary Note 4. The sentence reads now: An asymmetric slab consisting of seven W layers and two Fe layers was used as structural model.

Reviewer #3 This manuscript presents a theoretical proposal of antiskyrmions stabilized by anisotropic Dzyaloshinskii-Moriya interaction. Skyrmion has been a very

hot topic in the field of condensed matters since its first experimental observation in magnetic materials in 2009. This manuscript deals with the antiskyrmions which have been explored in this literature [e.g., Nature Commun. 7, 10542 (2016) etc.]. Nevertheless, Ref. [19] pointed out that antiskyrmions are unstable in a more symmetric system, whereas the present manuscript have carried out systematic theoretical and numerical studies of antiskyrmion stabilities in chiral magnets. This might be an important research advance of antiskyrmions since it has some novel properties as compared to the conventional skyrmions which have been intensively studied in the literature.

Reviewer #3 For example, it may add a new dimension and new degree of freedom in skyrmionic devices, such as manipulating its helicity etc for spin logic applications [see, Scientific Reports 5, 9400 (2015)].

Reply: Thank you for pointing this out to us. We have included this paper and we will come back to it below.

Reviewer #3 In a recent arxiv paper on frustrated skyrmion [arXiv:1703.07501], it is shown that frustrated antiskyrmions is stable solution and the antiskyrmion can interact with skyrmion in a fascinating way. So it would be interesting to compare this manuscript with the arxiv paper briefly.

Reply: We think that frustrated skyrmions are actually also a very interesting subject, but are distinctly different than the skyrmions in chiral magnets. In the absence of spin-orbit interaction and the magnetostatic self-energy or magnetic dipolar energy, respectively, Bloch and Néel type skyrmions and antiskyrmions of different helicity and chirality, respectively, have all the same energy, but some degeneracies are lifted at the presence of dipole-dipole energy, e.g. skyrmions and antiskrymions, Bloch-type skyrmions are favored over Néel-type skyrmions, but the degeneracies such as the helicity remains. Thus, the energy scale lifting degeneracies and the scale of the related energy barriers are of magnetostatic nature and relatively small with respect to chiral magnets, because here, the important energy scale lifting degeneracies is the Dzyaloshinskii-Moriya interaction. Both has advantages and disadvantages with respect to switching and dynamics. Obviously, in our case, the switching of helicity, which in our case would be the switching of chirality, as we deal here with Néel-type skyrmions, is not possible. On the other hand, skyrmions and antiskrymions differ in the magnetostatics. This little energy difference changes the current induced dynamics of skyrmions and antiskyrmions. This is different for anisotropic rank-two DMI materials whose DMI may be tuned such that by including the magnetostatic energy contribution the final result will be a rank-one DMI material where skyrmions and antiskyrmions can be used as skyrmions-antiskyrmion racetrack memory or in skyrmions logic.

We are afraid that the differences in the energy scales and in the degeneracies between the two types of skyrmions are so different that this would be worth being subject of a new paper and that this goes beyond what can be managed in this paper. On the other we acknowledge that we should have paid attention to the magnetostatic energy that we ignored basically. Thus, in summary we responded as following:

- (1) In the section “Results”, paragraph 3, we explain that the magnetization field of a skyrmion and an antiskrymion is different by a quadrupolar field component. Implicitly it is clear that the magnetostatic self-energy of skyrmions and antiskrymion embedded into a ferromagnetic background is different. This aspect we formulated explicitly introducing a new paragraph 6 in section “results”:

Magnetostatic energy difference between skyrmion and antiskyrmion, in which we also cite an additional reference [25], added a new Supplementary Note 3 -- Magnetostatic energy of axisymmetric skyrmions vs. antiskyrmions (all previous Supplementary Notes beyond Note 2 changed their number n to $n+1$) and added the following paragraph in the section "Discussion" of the main paper: The magnetostatic energy is not able to change the handedness of the skyrmion, but can be used to tune the energy difference between the skyrmion and antiskyrmion on a fine energy scale as function of the film thickness.

In total, we found a very interesting universal result in the way the magnetostatic energy contributes to Bloch- and Néel-type skyrmions as well as antiskyrmions, which adds to the value of the paper.

- (2) We added the reference suggested by the referee to the following sentence: This opens an exciting perspective to investigate the interaction of particle and antiparticle of different energies in the spirit of the skyrmion to antiskyrmion interaction in dipolar magnets [20] or frustrated magnets [44] and the conditions of mixed ordered lattices and phases.

Reviewer #3 In addition, I hope the authors can provide a brief discussion on the practical application of antiskyrmions, which might be beneficial for a wide readership.

Reply: We think this is a really good suggestion. We have divided the last section of the main paper "Discussion" into three separate ones and added a new last paragraph:

Rank-one materials should be particularly interesting for information storage and processing. One may envisage a creation process of a skyrmion-antiskyrmion pair out of the trivial FM state or a domain wall keeping the total \mathbb{S}^1 and \mathbb{S}^2 winding numbers, \mathbb{S}^1 and \mathbb{S}^2 , zero. They allow for an extension of the skyrmion race track idea~\cite{Fert_13}, where the information is encoded in the relative positions or time sequences, respectively, of the skyrmions along the track, to the skyrmion-antiskyrmion race track memory~\cite{Hoffmann_17}, where the binary information is encoded in the sequence of skyrmions and antiskyrmions, which are expected to be read out distinctly. Upon further investigations of skyrmion-antiskyrmion interactions in constrictions, rank-one materials may be the ideal host to extend the concepts of the skyrmion logic gates~\cite{Ezawa_15} to magnetic logic gates in which skyrmions and antiskyrmions are the elementary particles for binary operations.

Reviewer #3 In my view, the main results of interface stabilized antiskyrmions will contribute to the skyrmionic research by adding new dimension to this exciting field. It presents something very important and will motivate further research efforts towards its experimental realization. Therefore, I support the publication of this paper in Nature Communications provided the above comments can be addressed.

REVIEWERS' COMMENTS:

Reviewer #2 (Remarks to the Author):

I am satisfied with the response that the authors have provided to my earlier remarks. On second reading, I have noticed that the criterion for the stability of antiskyrmions (sign of $\det(D)$) is contained in the analysis of Ref. 22, which has a classification of 2D systems in their section II. In particular, see the discussion on page 3 and the caption of Fig. 2 of Ref. 22, where the inversion matrix changes the signs of $\det(D)$ and of the topological charge. This should be mentioned. I recommend this paper for publication once this issue has been addressed.

Reviewer #3 (Remarks to the Author):

I think the authors have satisfactorily addressed my comments. I would like to recommend this work to be published in Nature Communications.